# ALIGNMENT UNLOCKS COMPLEMENTARITY: A FRAMEWORK FOR MULTIVIEW CIRCUIT REPRESENTATION LEARNING

## ABSTRACT

Multiview learning on Boolean circuits holds immense promise, as different graph-based representations offer complementary structural and semantic information. However, the vast structural heterogeneity between views—such as an And-Inverter Graph (AIG) versus an XOR-Majority Graph (XMG)—poses a critical barrier to effective fusion, especially for self-supervised techniques like masked modeling. Naively applying such methods fails, as the cross-view context is perceived as noise. Our key insight is that *functional alignment is a necessary precondition to unlock the power of multiview self-supervision*. We introduce **MixGate**, a framework built on a principled training curriculum that first teaches the model a shared, function-aware representation space via an *Equivalence Alignment Loss*. Only then do we introduce a multiview masked modeling objective, which can now leverage the aligned views as a rich, complementary signal. Extensive experiments, including a crucial ablation study, demonstrate that our alignment-first strategy transforms masked modeling from an ineffective technique into a powerful performance driver.

## 1 INTRODUCTION

Multiview learning on Boolean circuits holds immense promise, as different graph-based representations offer complementary structural and semantic insights. While an And-Inverter Graph (AIG) provides a detailed structural view, a format like an XOR-Majority Graph (XMG) offers a semantically richer, high-level abstraction. This multiview approach has shown remarkable empirical success, surpassing earlier models that relied on single representations Li et al. (2022); Wang et al. (2022); Wu et al. (2023); Shi et al. (2023); Deng et al. (2024); Wang et al. (2024). The key challenge, however, arises from the vast structural heterogeneity between these views. This disparity poses a critical barrier to advanced self-supervised techniques like Masked Circuit Modeling (MCM) Shi et al. (2025b); Wu et al. (2025), which is inspired by the success of masked language modeling in natural language processing (NLP) Devlin et al. (2019). When a model lacks a common frame of reference, the cross-view context is perceived as noise rather than a useful signal, rendering such techniques ineffective.

Our key insight is that *fine-grained functional alignment is a necessary precondition to unlock the power of multiview self-supervision.* We argue that before a model can leverage complementary views for complex reasoning, it must first be guided to learn a shared, function-aware representation space. This alignment acts as a "Rosetta Stone", teaching the model to recognize that structurally alien subgraphs can be functionally equivalent, thereby bridging the gap between the different circuit "languages".

Building on this principle, we introduce **MixGate**, a framework designed around an alignment-first training curriculum (see Figure 1). MixGate's core is an **Equivalence Alignment Loss** that explicitly enforces functional consistency for these equivalent nodes across various views, building the shared representation space needed for effective fusion. Only after this foundation is established does our framework leverage a multiview masked modeling objective, transforming the now-aligned views into a rich, complementary signal for robust self-supervised learning. In our experiments, we choose three easily obtainable complementary views, Majority-Inverter Graph (MIG), XOR-AND Graph (XAG) and XOR-Majority Graph (XMG). Our results show significant performance

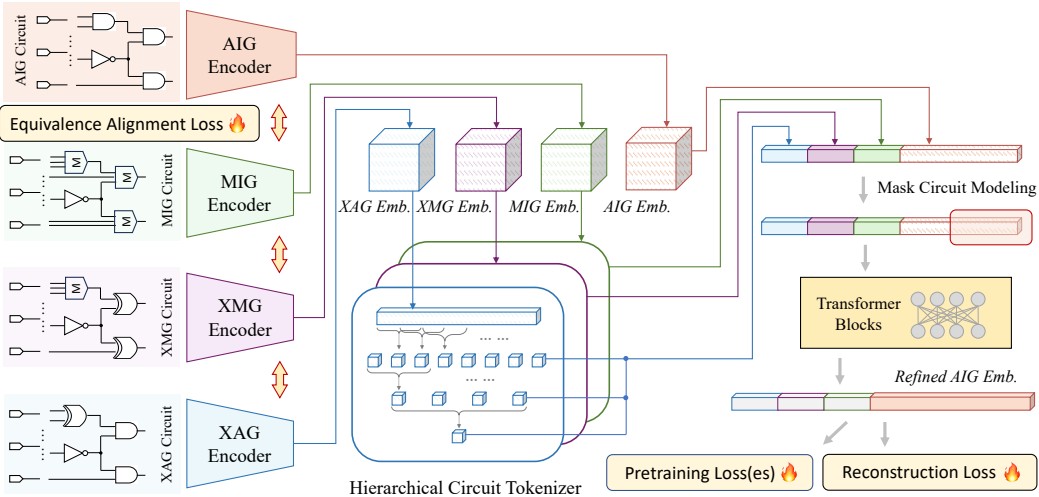

Figure 1: **The MixGate Framework**. A target circuit (e.g., And-Inverter Graph, AIG) is converted into multiple complementary views: Majority-Inverter Graph (MIG), XOR-AND Graph (XAG) and XOR-Majority Graph (XMG). Dedicated *graph encoders* process each view, and a *hierarchical circuit tokenizer* efficiently organizes the resulting embeddings into structured tokens. These multi-view tokens are fused by *Transformer blocks* to produce a refined output embedding, which is enriched with complementary features and leads to improved performance on downstream tasks

improvements, highlighting that establishing functional alignment is indeed the critical precondition for unlocking the full potential of multiview self-supervision in circuit modeling.

The main contributions of this work are summarized as follows:

- **Principled Solution to Multiview Heterogeneity:** We identify structural heterogeneity as a critical barrier for self-supervised learning on circuits and propose a novel alignment-first curriculum as solution. Analysis of the resulting model validates our approach, revealing an emergent attention behavior that naturally prioritizes the functionally aligned logic.

- **The MixGate Framework:** We present **MixGate**, a complete and effective framework embodying our alignment-first principle. MixGate integrates a novel hierarchical tokenizer for efficiency and is the first to successfully leverage multiview masked modeling by conditioning it on a pre-aligned representation space.

- **Comprehensive Empirical Validation:** We demonstrate through extensive experiments that our alignment-first strategy is critical for performance. Our ablation study proves that alignment **unlocks the potential of masked modeling**, turning a previously ineffective technique into a significant performance driver. Furthermore, we show that MixGate is a generalizable enhancement for a wide range of existing models.

## 2 RELATED WORK

### 2.1 GRAPH REPRESENTATIONS OF BOOLEAN CIRCUITS

A Boolean function can be implemented in various representations that leverage different sets of logic gates. Prominent formats include And-Inverter Graphs (AIGs) Mishchenko et al. (2006), Majority-Inverter Graphs (MIGs) Amarú et al. (2014), XOR-And Graphs (XAGs) Háleček et al. (2017), and XOR-Majority Graphs (XMGs) Haaswijk et al. (2017). For instance, in an AIG representation, logic circuits are modeled as directed acyclic graphs where AND and NOT gates are represented as nodes, and the wires connecting them are represented as directed edges.

All these representations preserve the functional equivalence of the Boolean circuit but vary in how effectively they support downstream tasks. AIGs are widely adopted in industrial tools due to their structural simplicity Barbareschi et al. (2022). MIGs generalize AIGs by replacing AND with majority logic, allowing delay-oriented optimization and more compact encoding of arithmetic and control circuits Amaru et al. (2015). XAGs extend AIGs with explicit XOR gates. They provide

more efficient synthesis for parity and arithmetic logic where XOR operations are prevalent Háleček et al. (2017). XMGs combine the strengths of MIGs and XAGs by including both XOR and majority gates, offering improved area-delay trade-offs and a balanced representation of control and datapath logic Chu et al. (2019).

Recent research emphasizes the complementary nature of circuit representations within unified logic optimization frameworks Neto et al. (2019); Pu et al. (2025); Fu et al. (2025). For instance, LSO-racle Neto et al. (2019) partitions circuits into clusters. Each cluster is converted into the most suitable representation among AIG, MIG, XAG, or XMG, and processed by dedicated optimization techniques. Experimental results show that this heterogeneous strategy leads to improved power, performance, and area (PPA) metrics in the final synthesized circuits. Our work aims to leverage this complementarity in circuit representation learning, and further enhance model performance by learning multiview information across these diverse graph formats.

## 2.2 Circuit Representation Learning

Circuit representation learning Li et al. (2022); Shi et al. (2023); Wang et al. (2022) employs deep learning models to extract informative and general-purpose embeddings of Boolean circuits. These embeddings encode both structural and functional properties and have demonstrated good performance across various EDA tasks, such as testability analysis Shi et al. (2022) and Boolean reasoning Wu et al. (2023).

Given the inherently multimodal nature of the EDA design flow, circuit designs can be represented at various abstraction levels, such as hardware description language (HDL) code, gate-level netlists, and physical layouts. Recent studies have explored fusing information across these modalities to improve downstream task performance Chen et al. (2024); Zhong et al. (2024); Shi et al. (2025b); Fang et al. (2025); Wu et al. (2025). Generally, mask modeling is a promising self-supervised approach to fuse the cross-view information, which is inspired by BERT Devlin et al. (2019). However, while effective in NLP applications and single-view settings, the application of mask modeling to the multiview circuit learning domain, with its inherent structural heterogeneity, remains an open problem. For example, the vast structural differences between views like AIGs and XMGs mean that cross-view context is often perceived as noise rather than a useful signal, rendering naive applications of masked modeling ineffective. Our work addresses this gap by proposing an alignment-first curriculum as a necessary precondition for effective multiview self-supervision.

## 3 MixGate Framework

As motivated in the introduction, the core challenge is that multiview circuit graphs (e.g., AIG, XMG, MIG, XAG) exhibit vast structural heterogeneity. A successful framework must therefore unify these views into a common embedding space while preserving their complementary functional and structural information. To this end, **MixGate** integrates three essential components: (1) systematic preparation of multiview data with fine-grained correspondences, (2) a hierarchical tokenizer and Transformer backbone for fusion, and (3) a progressive alignment-first training strategy.

### 3.1 Data Preparation

**Dataset Source** We construct our dataset for MixGate training based on the open-source ForgeEDA dataset Shi et al. (2025a), which provides 1,189 large-scale and high-quality circuits across 20 divisions. The description of raw data is summarized in Appendix A.1.

**Multiview Data Construction** The MIG, XMG, and XAG are generated by converting AIG using ALSO [1], an open-source logic synthesis tool. For each input AIG, the command `lut_mapping` is first utilized to transform the AIG into a Look-up Table (LUT) network consisting of 4-input LUTs (4-LUTs). Subsequently, the 4-LUT network is converted to the corresponding MIG, XMG, and XAG through the command `lut_resyn`. The entire transformation process is computationally efficient, with approximately linear time complexity in relation to circuit size. Despite the different views, these graphs of the same Boolean circuit have the same function. Detailed flows are provided in Appendix A.2.

---

[1] ALSO: Advanced Logic Synthesis and Optimization tool. https://github.com/nbulsi/also

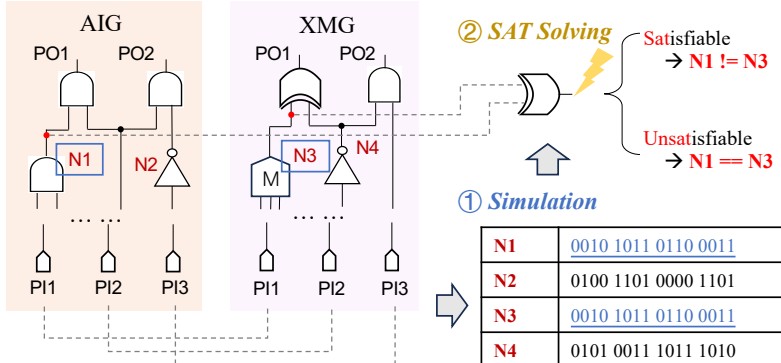

Figure 2: Example of equivalence gates identification

**Equivalence Gates Identification**  We identify functionally equivalent nodes across different views to establish the ground truth for equivalence alignment. The label preparation process is inspired by SAT sweeping Kuehlmann et al. (2002), a well-established technique to merge equivalence pairs for area minimization in logic synthesis. As shown in Figure 2, we first perform random *simulation* on the different circuit graphs of the same design and figure out these nodes with the same response. This step serves as a fast and efficient filter, drastically reducing the number of candidate pairs by eliminating obviously non-equivalent nodes. Second, for each candidate pair, we construct a miter circuit by feeding their outputs into an XOR gate and formally check equivalence using a SAT solver. If the solver returns an UNSAT (unsatisfiable) result, it formally proves that no input vector exists for which the nodes' functions differ, confirming their functional equivalence. Otherwise, these two nodes are not equivalent. If node N1 from AIG and node N3 from XMG always share the same Boolean function, we annotate these two nodes as equivalent nodes and enforce the proposed model produces the same embeddings for these two nodes.

Finally, we construct multiple graph-based representations (AIG, MIG, XAG, and XMG) for each circuit design and enriched them with fine-grained annotations of functionally equivalent nodes. We will release both our dataset and code to contribute to the open-source EDA and AI community.

## 3.2 MODEL ARCHITECTURE

Figure 1 illustrates the modular architecture of the proposed MixGate framework, which exemplifies how to refine the embeddings of an AIG netlist by integrating multiview circuit information. The framework consists of four *graph encoders*, a novel hierarchical *circuit tokenzier*, and plain *Transformer blocks*.

**Graph Encoder**  We design four specialized graph encoders tailored to AIG, MIG, XAG, and XMG, respectively. These encoders operate on their respective netlists and are crafted to obtain the structural embeddings $hs$ and functional embeddings $hf$ in each view. The encoders are defined as below:

$$H_v^S, H_v^F = E_v(G_v), \quad H_v^S = [hs_1^v, hs_2^v, \ldots, hs_n^v], \quad H_v^F = [hf_1^v, hf_2^v, \ldots, hf_n^v]. \tag{1}$$

where $E_v$ represents the encoder for view $v \in \{\text{AIG}, \text{MIG}, \text{XAG}, \text{XMG}\}$, and $G_v$ represents the input circuit graph. The four view-specific encoders are first pretrained separately and then jointly fine-tuned within MixGate to remain fully trainable during end-to-end optimization.

Within the graph encoder, we extend the directed acyclic graph-aware aggregator proposed in Deep-Gate2 Shi et al. (2023), which is elaborated in Appendix B. Briefly, for a node $k$ with gate type $type(k)$, let $\mathcal{P}(k)$ denote its set of predecessor nodes. We adopt the aggregator to obtain the functional embedding $hf$ and structural embedding $hs$ of node $k$. As shown in Eq. 2, the trainable aggregator in graph encoders is defined as $\Phi = \{\phi_{type(k)}^s \phi_{type(k)}^f\}$. All the node embeddings are updated from Primary Inputs (PIs) to Primary Outputs (POs) level by level, which mimics the behavior of Boolean logic computation.

$$hs_k = \phi_{type(k)}^s(hs_i | i \in \mathcal{P}(k)), \quad hf_k = \phi_{type(k)}^f([hf_i, hs_i] | i \in \mathcal{P}(k)). \tag{2}$$

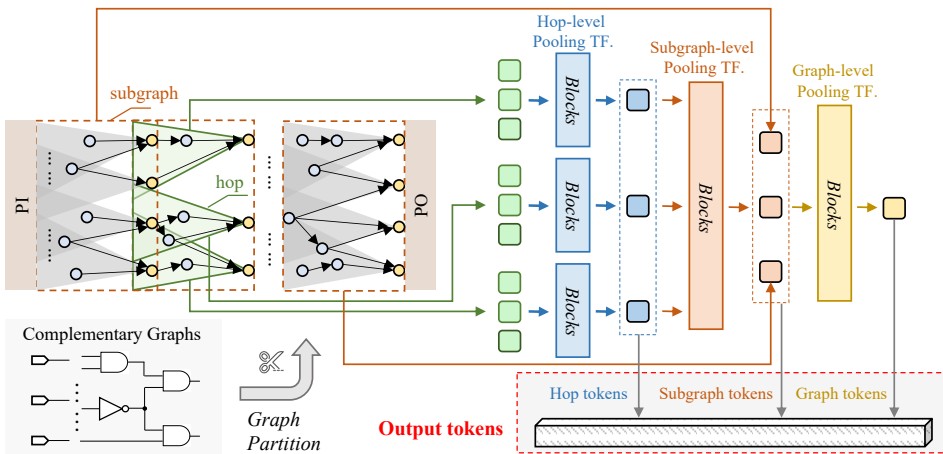

Figure 3: Overview of hierarchical circuit tokenizer

Then, we distinguish the aggregator according to the gate type $type(k)$. For AIG encoding, the encoder $E_{AIG}$ contains two aggregators: $E_{\text{AIG}} = \{\Phi_{\text{AND}}, \Phi_{\text{NOT}}\}$.

An MAJ gate can be degraded into a simpler gate when one of its inputs is a constant (0/1). For example, if one of the inputs is a constant 0, the gate behaves like an AND gate. If the input is a constant 1, the gate acts as an OR gate. This degradation can be formally described as:

$$\text{MAJ}(A, B, C) = \begin{cases} \text{AND}(A, B), & \text{if } C = 0, \\ \text{OR}(A, B), & \text{if } C = 1, \\ \text{MAJ}(A, B, C), & \text{otherwise.} \end{cases} \quad (3)$$

Such behavior implicitly alters gate functionality during simulation, which may mislead the aggregation process in GNNs. To address this, we explicitly differentiate between native and degraded forms of the MAJ gate. Specifically, we extend the set of gate-specific aggregators to include degraded forms, introducing AND and OR gates for the graph views with MAJ gate. Accordingly, we define the encoder sets $E$ for the other three views as: $E_{\text{MIG}} = \{\Phi_{\text{MAJ}}, \Phi_{\text{AND}}, \Phi_{\text{OR}}, \Phi_{\text{NOT}}\}$, $E_{\text{XAG}} = \{\Phi_{\text{AND}}, \Phi_{\text{XOR}}, \Phi_{\text{NOT}}\}$, $E_{\text{XMG}} = \{\Phi_{\text{XOR}}, \Phi_{\text{MAJ}}, \Phi_{\text{AND}}, \Phi_{\text{OR}}, \Phi_{\text{NOT}}\}$.

**Circuit Tokenizer**  Circuit tokenizer $\theta$ transforms the embeddings into sequences of tokens, the input to the Transformer blocks for further processing. For the refined view, such as AIG in Figure 1, we use a flat tokenization strategy, directly treating its node-level embeddings as the input sequence without hierarchical grouping: $[t_1, t_2, \cdots, t_n]_v = \theta(H_v^S, H_v^F)$, $\mathbf{T}_v = [t_1, t_2, \cdots, t_n]_v$.

**Hierarchical Circuit Tokenizer**  For the complementary graphs (MIG, XAG and XMG), directly passing them to a Transformer risks flattening important structural cues and leads heavy computational complexity. To address this, we introduce a hierarchical tokenizer that aggregates node embeddings into hop-level, subgraph-level, and graph-level tokens, thereby extracting multiple level information and significantly reducing the number of tokens. The proposed tokenizer operates in a three-level hierarchy: *hop-level*, *subgraph-level*, and *graph-level* to orchestrate the final *output tokens*.

First, as shown in Figure 3, given an input graph $G = (\mathcal{V}, \mathcal{E})$, where $\mathcal{V}$ is the set of nodes and $\mathcal{E}$ is the set of edges, graph partitioning begins from the Primary Outputs (POs) and proceeds backward into the Primary Inputs (PIs) in discrete steps. Each logic hop $\mathcal{H}_i$ consists of nodes and edges reachable within a fixed fan-in depth of $l$ levels. For example, the first hop $\mathcal{H}_1$ contains one of the POs and the logic gates in their fan-in up to $l$ levels deep. The subsequent hops begin at the gates in $d - l$ level and extend a further $l$ levels back toward the PIs, where $d$ is the total depth of the given graph. This creates a series of sequential and non-overlapping $N_h$ hops that segment the entire graph from output to input.

Next, we group the hops into larger subgraphs $\mathcal{S}_j$ by aggregating those that belong to the same level according to a pre-defined index set $\mathbf{L}_j$. Each subgraph is formed as $\mathcal{S}_j = \bigcup_{k \in \mathbf{L}_j} \mathcal{H}_k, \forall j \in$

$\{1, \ldots, N_s\}$, where $\mathbf{L}_j$ denotes the hop indices included in $\mathcal{S}_j$ and $N_s$ is the total number of sub-graphs. After constructing all subgraphs, we combine them to obtain the full graph $\mathcal{G}$, defined as $\mathcal{G} = \bigcup_{m=1}^{N_s} \mathcal{S}_m$.

The above bottom-up hierarchical decomposition reorganizes the circuit graph into three progressively coarser levels: hop-level, subgraph-level, and graph-level, enabling multiscale structural representation for downstream processing. Given the node-level embeddings within a hop area $t_k$, $k \in \mathcal{H}_i$, we generate the hop-level token $t_{\mathcal{H}_i}$ by pooling the corresponding node embeddings. To be specific, we adopt a pooling Transformer (Pooling TF.), which prepends a learnable special token [CLS] to the input sequence. This special token attends to all other tokens in the sequence, and its output representation serves as a summary. The pooling process is formalized as: $[\text{CLS}]' = \delta([\text{CLS}], t_1, t_2, \cdots, t_n)$, where the output $[\text{CLS}]'$ is the pooling result.

Therefore, the hop-level token is obtained by aggregating the contained nodes, i.e., $t_{\mathcal{H}_i} = \delta(t_k)$, $k \in \mathcal{H}i$. Similarly, the subgraph-level token is computed by pooling over its hop tokens as $t_{\mathcal{S}j} = \delta(t_{\mathcal{H}_i})$, $\mathcal{H}i \in \mathcal{S}j$, and the single graph-level token is produced by pooling over all subgraph tokens, $t_{\mathcal{G}} = \delta(t_{\mathcal{S}_j})$, $\mathcal{S}_j \in \mathcal{G}$.

The final output of our hierarchical tokenizer is the concatenation of all generated tokens across the three levels: $\mathbf{T} = \{t_{\mathcal{H}_i}\}_{i=1}^{N_h} \cup \{t_{\mathcal{S}_j}\}_{j=1}^{N_s} \cup t_{\mathcal{G}}$.

**Transformer Blocks**  To effectively fuse multiview information of various Boolean circuit graphs, we use the Transformer blocks in MixGate framework. Formally, the Transformer blocks $\pi$ are denoted as: $\mathbf{T}'_{\text{AIG}}, \mathbf{T}'_{\text{MIG}}, \mathbf{T}'_{\text{XAG}}, \mathbf{T}'_{\text{XMG}} = \pi(\mathbf{T}_{\text{AIG}}, \mathbf{T}_{\text{MIG}}, \mathbf{T}_{\text{XAG}}, \mathbf{T}_{\text{XMG}})$. Inspired by the sparse Transformer architecture in Zheng et al. (2025), we implement the multi-head attention mechanism in the traditional Transformer with a graph attention network (GAT) to improve the efficiency.

### 3.3 MODEL TRAINING

As outlined in Figure 1, our strategy rests on two pillars: (1) establishing functional alignment across structurally diverse views, and (2) exploiting their complementary context via multiview fusion. To operationalize these, MixGate employs three types of loss functions.

**Equivalence Alignment Loss**  To harness the complementary information across different circuit representations (AIG, XMG, XAG, MIG), we explicitly enforce functional consistency among them during training. We introduce an **Equivalence Alignment Loss** ($L_{align}$) to pull the embeddings of these equivalent nodes closer together in the latent space. Formally, for a pair of nodes $(i, j)$ from two different views $i \in G_{v_1}, j \in G_{v_2}$ with the same Boolean behavior, we minimize the L1 distance between their functional embeddings $hf$ produced by graph encoders.

$$L_{align} = \frac{1}{|\mathcal{P}|} \sum_{(i,j) \in \mathcal{P}} ||hf_i - hf_j||_1 \tag{4}$$

where $\mathcal{P}$ is the set of all functionally equivalent node pairs identified across the multiview circuits. In practice, $\mathcal{P}$ can be subsampled per batch to reduce computational overhead without degrading alignment quality. We adopt the L1 distance for its simplicity and robustness. It provides a stable anchor for positive pairs in highly heterogeneous circuit graphs, while contrastive objectives with negative sampling add substantial complexity without clear benefits. Further justification and empirical analysis are provided in Appendix H.

**Reconstruction Loss**  Once a unified embedding space is established, the model can now benefit from self-supervised signals across views. We therefore extend masked modeling into a multiview setting, where a masked cone in one graph must be reconstructed not only from intra-view cues but also from complementary cross-view context.

For a target view (e.g., AIG, $\mathcal{G}^A$), we randomly select a node $p$ and mask its entire $k$-hop input cone $\mathcal{M}(p)$, as illustrated in Figure 4. The functional embeddings $hf_i$ of all nodes $i \in \mathcal{M}(p)$ are replaced by a learnable mask token $hm$, while their structural embeddings $hs_i$ are preserved. The models are then fed with the combined tokens from all views: the masked AIG tokens, the unmasked tokens from the other source views (XMG, XAG, MIG, $\mathcal{G}^S$), and the unmasked tokens from the rest of the AIG: $\mathbf{T}^*_{\text{AIG}} = \text{Mask}(\mathbf{T}_{\text{AIG}})$.

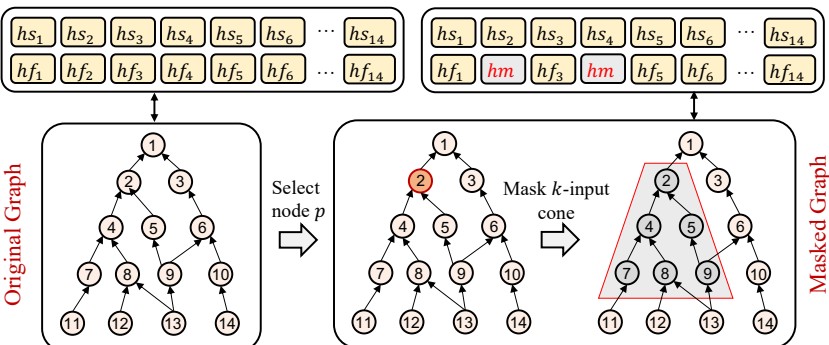

Figure 4: Example of mask circuit modeling

The combined sequence of tokens $\mathbf{T}^*_{\text{AIG}}, \mathbf{T}_{\text{MIG}}, \mathbf{T}_{\text{XAG}}, \mathbf{T}_{\text{XMG}}$ is processed by the Transformer encoder $\pi$ to reconstruct contextualized embeddings. The training objective is to reconstruct the original functional embeddings of the masked nodes using a linear decoder head and an L1 loss:

$$L_{mcm} = \frac{1}{|\mathcal{M}(p)|} \sum_{i \in \mathcal{M}(p)} ||\mathbf{T}^{'}_{\text{AIG}} - \mathbf{T}_{\text{AIG}}||_1 \tag{5}$$

**Pretraining Losses** As prior works Shi et al. (2023); Liu et al. (2024); Shi et al. (2024), we read out the node-level tokens for the following pretraining tasks. These two loss metrics are used to evaluate model performance in the next section. For both metrics, smaller values indicate better performance. More detailed definition of these two loss functions can be found in Appendix C.

- **Signal Probability Prediction (SPP) Loss** ($L_{spp}$): Predict the probability that a logic gate outputs a logical 1 under random input simulation. It is computed as the average absolute difference between the predicted and ground-truth signal probabilities. SPP is particularly relevant for testability analysis Williams & Angell (1973).
- **Truth-Table Distance Prediction (TTDP) Loss** ($L_{ttdp}$): Measure the semantic distance between pairs of logic gates by predicting the difference in their truth tables. It is calculated as the average absolute difference between predicted and ground-truth distances based on random simulation. TTDP is important in logic synthesis Kuehlmann et al. (2002), as a distance of zero implies functional equivalence, enabling gate merging.

**Training Pipeline Overview** We consolidate the training into the **MixGate Curriculum**. Rather than applying all training objectives simultaneously, we first align the latent spaces of graph encoders and then fuse multiview information by mask modeling. A three-stage progression is adopted to stabilize optimization (details in Appendix C).

## 4 EXPERIMENTAL RESULTS

In this section, we present experiments designed to validate the central claim of this paper: *functional alignment is the prerequisite that enables multiview self-supervision to succeed in circuit representation learning*. We begin with a crucial ablation study that directly supports this claim (Sec. 4.1) and then explore the optimal hyperparameters (Sec. 4.2). In the following ablation studies, we investigate the effects of multiview circuit learning in Sec. 4.3 and the novel hierarchical circuit tokenizer in Sec. 4.4. Finally, we demonstrate that MixGate framework can be generalized to various circuit encoders in App. D and large-scale circuits in App. E. We then inspect the contributions of each view (see App. F) in the appendices. The model implementation details with default hyperparameters are elaborated in App. G.

### 4.1 ALIGNMENT AS THE PRECONDITION FOR MULTIVIEW SELF-SUPERVISION

To evaluate the effectiveness of the proposed fine-grained equivalence alignment mechanisms, we conduct an ablation study with four variants: (1) **Baseline**, the baseline without mask or alignment, (2) **+Mask**, which incorporates only the masked circuit modeling objective, (3) **+Align**,

Table 1: Ablation on equivalence alignment

| Variants | Loss Values | | | | | |
|---|---|---|---|---|---|---|
| | $L_{spp}$ | Red. ($\downarrow$) | $L_{ttdp}$ | Red. ($\downarrow$) | $L_{mcm}$ | $L_{align}$ |
| Baseline | 0.0242 | | 0.0837 | | / | / |
| +Mask | 0.0247 | -2.02% | 0.0896 | -6.58% | 0.1770 | / |
| +Align | 0.0236 | 2.54% | 0.0828 | 1.09% | / | 0.0710 |
| +Mask +Align | **0.0226** | **7.08%** | **0.0797** | **5.02%** | **0.1690** | **0.0637** |

which applies only the node-level functional alignment constraint across various views, and (4) **+Mask+Align** combines both mechanisms, which is the proposed MixGate settings. All the variants share the model framework with the same number of trainable parameters.

The results are shown in Table 1. First, using mask alone (**+Mask**) is counterproductive, leading to a noticeable performance degradation compared to the baseline. For instance, the $L_{spp}$ loss worsens from 0.0242 to 0.0247 (a 2.02% improvement) and $L_{ttdp}$ loss increases by 6.58%. Our results indicate that without alignment, masked modeling in isolation fails, as the structurally diverse cross-view context is treated as noise rather than a useful signal.

Second, using align alone (**+Align**) consistently improves performance by enforcing functional consistency across the different circuit views. The alignment reduces the $L_{spp}$ loss to 0.0236 (a 2.54% reduction) and the $L_{ttdp}$ loss to 0.0828 (a 1.09% reduction).

Third, combining both mechanisms (**+Mask +Align** setting) yields the best results by a significant margin, which achieves the lowest losses, reducing $L_{spp}$ by 7.08% and $L_{ttdp}$ by 5.02% compared to the baseline without incorporating any inference overhead. This ablation study highlights that alignment is essential for realizing the benefits of multiview self-supervision. In the following experiments, we choose the best setting (**+Mask +Align**) for further investigations.

## 4.2 EXPLORATION ON MASK RATIO

We further analyze the impact of different mask ratios in the self-supervised MCM training strategy. The mask ratio is defined as the percentage of nodes in a circuit that are randomly selected to have their input cones masked. It should be denoted that since each selection masks an entire input cone, the total portion of the masked circuit is significantly larger than the mask ratio. We vary the mask ratio from **0.00** to **0.05**, where the **0.00** mask ratio is identical to the **+Align** setting without masking.

Table 2: Effect of different mask ratios

| Mask Ratio | Loss Values | | | |
|---|---|---|---|---|
| | $L_{spp}$ | $L_{ttdp}$ | $L_{mcm}$ | $L_{align}$ |
| 0.00 | 0.0236 | 0.0828 | / | 0.0710 |
| 0.01 | 0.0231 | 0.0812 | 0.1528 | 0.0996 |
| **0.03** | **0.0226** | **0.0797** | **0.1693** | **0.0654** |
| 0.05 | 0.0269 | 0.0984 | 0.1032 | 0.0594 |

From Table 2, the results confirms that a small amount of masking is better than none. Starting from the **0.00** ratio baseline, increasing the mask ratio to **0.01** and **0.03** progressively improves performance. The model achieves its best results on SPP and TTDP at the **0.03** mask ratio, with $L_{spp}$ is 0.0226 and $L_{ttdp}$ is 0.0797. Besides, the experiment demonstrates that there is a tipping point. When the mask ratio is increased to **0.05**, performance sharply degrades, where the $L_{spp}$ rises to 0.0269 an $L_{ttdp}$ to 0.0984. Therefore, an excessively high mask ratio removes too much contextual information, overwhelming the model and hindering its ability to learn effectively.

## 4.3 IMPACT OF MULTIVIEW INFORMATION

To prove that the proposed multiview learning approach is robust and truly beneficial, we compare its performance against a traditional single-view setup. We create two distinct settings for refining each circuit types (AIG, MIG, XMG and XAG), respectively. Refining AIG embeddings with multiview information **(w/ Multiview)** is the default framework in this paper, which uses the corresponding MIG, XMG and XAG as auxiliary views. The encoders for all four views process their respective graphs, and the Transformer blocks fuse all this information to produce a refined AIG embedding. Without multiview (**w/o Multiview**) settings only have a single input graph. The learned embeddings

Table 3: Model performance of without (w/o) and with (w/) multiview

| Refined Graph | w/o Multiview | | w/ Multiview | | | |
|---|---|---|---|---|---|---|
| | $L_{spp}$ | $L_{ttdp}$ | $L_{spp}$ | Red. ($\downarrow$) | $L_{ttdp}$ | Red. ($\downarrow$) |
| AIG | 0.0247 | 0.1156 | 0.0226 | 8.50% | 0.0797 | 31.11% |
| MIG | 0.0323 | 0.0717 | **0.0284** | **12.07%** | 0.0431 | 39.89% |
| XAG | 0.0254 | 0.1206 | 0.0237 | 6.69% | **0.0686** | **43.16%** |
| XMG | 0.0235 | 0.0308 | 0.0217 | 7.67% | 0.0253 | 17.86% |

Table 4: Model performance of without (w/o) and with (w/) hierarchical circuit tokenzier

| Refined Graph | w/o Hierarchical Tokenzier | | | | w/ Hierarchical Tokenzier | | | |
|---|---|---|---|---|---|---|---|---|
| | $L_{spp}$ | $L_{ttdp}$ | Mem. (MB) | Time (s) | $L_{spp}$ | $L_{ttdp}$ | Mem. (MB) | Time (s) |
| AIG | 0.0221 | 0.0764 | 12043.89 | 13.79 | 0.0226 | 0.0797 | 8674.12 (23.56%$\downarrow$) | 7.16 (48.08%$\downarrow$) |
| MIG | 0.0289 | 0.0436 | 10782.38 | 12.41 | 0.0284 | 0.0431 | 7913.19 (27.98%$\downarrow$) | 6.97 (43.84%$\downarrow$) |
| XAG | 0.0241 | 0.0701 | 9618.22 | 12.08 | 0.0237 | 0.0686 | 7132.96 (25.84%$\downarrow$) | 6.35 (47.43%$\downarrow$) |
| XMG | 0.0220 | 0.0259 | 9431.14 | 11.75 | 0.0217 | 0.0253 | **6984.33 (25.94%$\downarrow$)** | **6.02 (48.77%$\downarrow$)** |

from graph encoder are also fed into the same Transformer blocks to ensure the identical parameter counts for fair comparison.

Column "Red." in Table 3 shows consistent gains from multiview fusion. The performance improvement is significant, where refining AIG embeddings using multiview information reduce TTDP loss $L_{ttdp}$ by a remarkable 31.11%. Notably, refining XAG with multiview inputs reduces SPP loss by 6.69% and TTDP loss by 43.16%. This confirms that MixGate effectively leverages complementary modalities to boost accuracy and adaptability—especially in tasks requiring cross-view integration.

### 4.4 IMPACT OF HIERARCHICAL CIRCUIT TOKENIZER

To evaluate the effectiveness of our proposed hierarchical circuit tokenizer, we compare its performance against a flat tokenization baseline across multiple circuit graphs. In the baseline setting (denoted as w/o Hierarchical Tokenizer), the input circuit embeddings are tokenized in a simple, sequential manner without structural hierarchy. All node-level embeddings produced by four graph encoders are considered as tokens and fed to the following Transformer blocks. In contrast, the w/ Hierarchical Tokenizer configuration employs our hierarchical tokenizer, which organizes embedding sequences based on the topological structure of the Boolean circuits, enabling the model to capture hop-to-graph patterns more effectively. We also record the average memory usage and inference time (including circuit transformation) on the validation set to assess computational efficiency. All models are trained with the same hyperparameters.

We draw two key observations from the Table 4. First, despite compressing the token space, the hierarchical tokenizer maintains comparable performance to the flat baseline. For instance, in the MIG refinement task, it achieves a slight improvement of 1.73% in SPP and a 1.15% reduction in TTDP, indicating that the hierarchical abstraction does not compromise functional accuracy. Second, due to the reduced token count, the hierarchical tokenizer substantially improves computational efficiency. For example, in the XMG case, it reduces memory usage by 25.94% and inference time by 48.77% compared to the flat baseline.

## 5 CONCLUSION

In this work, we address the key challenge of structural heterogeneity in multiview learning for Boolean circuits. We show that such heterogeneity renders powerful self-supervised techniques like masked modeling ineffective when applied directly. Our core principle is that functional alignment is a necessary prerequisite for unlocking the benefits of multiview self-supervision. Building on this insight, we design the MixGate framework with an alignment-first curriculum: the model is first guided to establish a shared semantic space through alignment, after which masked modeling can effectively exploit complementary signals. Experimental results demonstrate that our proposed strategy transforms masked circuit modeling from an ineffective objective into a strong performance driver. By establishing the primacy of alignment, this work provides a principled path toward more robust and effective self-supervised models for circuit representation learning.

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

# APPENDIX

## A   DATA PREPARATION

### A.1   OVERVIEW OF FORGEEDA DATASET

ForgeEDA Shi et al. (2025a) [2] is a comprehensive circuit dataset that consists of 1,189 practical circuit designs. The dataset is not generated by randomly crawling Verilog files from the Internet. Specifically, the major categories of Integrated Circuit (IC) products are first enumerated, such as arithmetic units, processors, encoders/decoders, and controllers. For each category, a curated set of domain-specific keywords is prepared. Using these targeted keyword queries, circuit designs are collected from various reliable sources across the Internet. As a result, ForgeEDA covers nearly all major classes of real-world IC designs, providing broad diversity and strong relevance for research in circuit learning and EDA.

To enable efficient training and ensure a fair comparison with existing methods, we adopt the same circuit partitioning strategy as used in prior works Shi et al. (2023; 2024); Liu et al. (2024). Specifically, we decompose the original designs into approximately 15,000 sub-circuits, which serve as individual samples for learning. These sub-circuits retain topological characteristics representative of their parent designs, making them suitable for evaluating representation learning models. The statistics of dataset is shown in Table 5.

Table 5: The statistics of dataset

|      | # Nodes | | | # Logic Levels | | |
| --- | --- | --- | --- | --- | --- | --- |
|      | Range | Avg. | Std. | Range | Avg. | Std. |
| AIG | [50, 1,499] | 931.69 | 290.17 | [8, 314] | 27.73 | 19.92 |
| MIG | [237, 1,786] | 924.83 | 334.73 | [6, 130] | 24.86 | 10.93 |
| XAG | [188, 1,678] | 929.56 | 306.53 | [9, 253] | 27.54 | 17.42 |
| XMG | [137, 1,590] | 827.82 | 325.27 | [6, 127] | 19.69 | 9.65 |

### A.2   PIPELINE OF DATASET PREPARATION

#### A.2.1   MULTIVIEW DATASETS CONSTRUCTION

To enable multiview learning across circuit representations, we construct datasets by converting AIG netlists into alternative graph formats including MIG, XAG, and XMG. This transformation is achieved using the ALSO logic synthesis tool, which preserves functional equivalence across different views while allowing graph representation diversity.

Figure 5 shows an example flow generating multiview graphs for model training and the annotated labels for attention analysis. Such process begins with an AIG, a directed acyclic graph where internal nodes represent 2-input AND gates and edges may carry inversion function (NOT gate). Using the `lut_mapping -k 4` command in ALSO, the AIG is first converted into a 4-input Look-Up Table (4-LUT) network. This intermediate LUT netlist serves as a unified functional representation, where each node computes a Boolean function defined by a truth table.

The second step applies the `lut_resyn` command to resynthesize each LUT into a specific graph format (e.g., AIG or XMG), using exact logic decomposition. Each LUT is processed in the topological order and replaced by an implementation of gates in the target representation (e.g., majority and XOR logic for XMG, or AND and NOT gates for AIG). This procedure ensures that although the resulting netlists have different internal structures, they remain functionally equivalent.

---

[2] ForgeEDA Dataset. https://github.com/cure-lab/LCM-Dataset

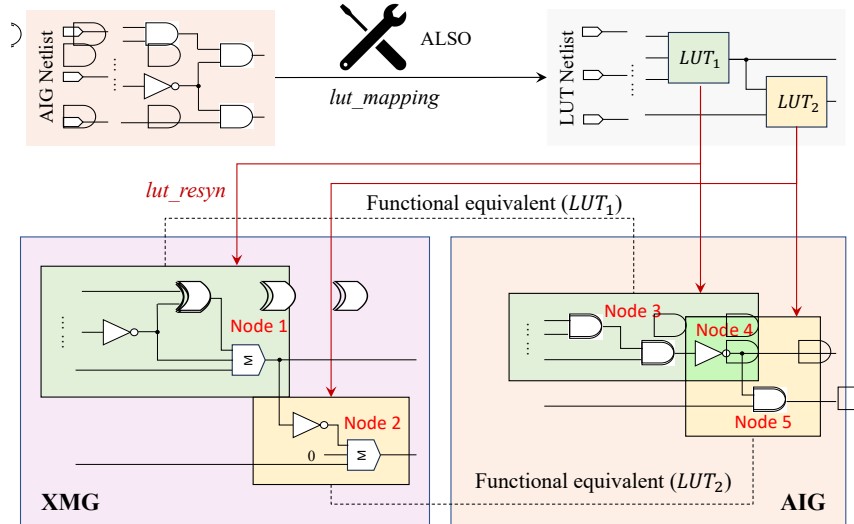

Figure 5: An example of multiview graphs (XMG and AIG) generation

### A.2.2 MULTIVIEW EQUIVALENT NODES CONSTRUCTION

To establish connections between circuits of different modalities and leverage their potential inter-relationships for complementary information exchange, we label the functionally equivalent nodes between the AIG and the other three circuit modalities (XMG, XAG, and MIG).

A technique for identifying functionally equivalent nodes is employed. We first transform the circuit from the AIG format into the another format (e.g. MIG, XAG and XMG). Figure 2 shows an example to identify the equivalence nodes between AIG and XMG views. Then, random simulation is performed on both circuits, hashing the resulting truth table for each node to generate a deterministic fingerprint for comparison. This methodology enables the swift elimination of nodes lacking functional equivalence, leading to a considerable reduction in temporal computational costs.

The next step involves applying SAT sweeping to two candidate nodes sharing the same key in the hash table. This technique is extensively employed in logic synthesis for equivalence checking, enabling efficient verification of functional equivalence between circuit nodes. We construct a XOR gate between these two nodes and set its output to logical 1. This forces the XOR to require the two fanins to differ, which can be used in the following SAT sweeping procedure.

We then convert the new circuit into conjunctive normal form (CNF) and perform SAT sweeping. If the SAT solver returns UNSAT, this indicates that no input assignment exists where the two nodes produce different values — thereby proving their functional equivalence. Otherwise, the solver find an assignment to differentiate these nodes. Such assignment is employed as another input pattern for incremental simulation to further filter out candidate nodes.

By locating the corresponding embeddings of functionally equivalent circuit nodes through indexing, we establish connections across different circuit modalities and minimize embedding discrepancies to achieve functional alignment.

## B AGGREGATOR OF GRAPH ENCODERS

In this section, we describe the self-attention mechanism and aggregators to capture both functional and structural information for each logic gate during one round of forward propagation.

Inspired by DeepGate2 Shi et al. (2023), our model separates functional embeddings $h_f$ and structural embeddings $h_s$ and initializes them differently. Functional embeddings are uniformly initialized for primary inputs (PIs) because they all share the same logic probability during random simulation. For structural embeddings, we employ a specialized PI encoding strategy. Each PI is

assigned a unique identifier as its initial structural embedding. Specifically, the initial structural embeddings $h_{si}, i \in PI$, are one-hot encoding vectors, which ensures that the dot product between any two PI embeddings is zero, meaning that they are independent of each other.

Each aggregator uses the self-attention mechanism. For a logic gate, its **controlling input** is an input value that determines the output of the gate regardless of the other inputs. For example, an AND gate outputs a logic 0 if any of its fan-in is a logic 0. The self-attention mechanism allows the model to give more weight to controlling inputs, enabling more accurate processing.

The attention coefficients, denoted as $\alpha_j$, are computed using the softmax function. These coefficients measure the importance of each predecessor node. The final aggregation function combines messages from all predecessor nodes, weighted by these attention coefficients. The aggregation process is formalized as

$$\alpha_j = \text{softmax}\left(\frac{w_q^\top h_i \cdot (w_k^\top h_j)^\top}{\sqrt{d}}\right), \tag{6}$$

where $w_q, w_k, w_v$ are weight matrices, and $d$ is the dimension of the embedding vectors. The softmax function is applied to the dot product between the query vector $w_q^\top h_i$ and the key vector $w_k^\top h_j$, scaled by $\sqrt{d}$, to calculate the attention coefficients. The final step in the aggregation is as follows:

$$h_i = \phi(h_j | j \in P(i)) = \sum_{j \in P(i)} (\alpha_j \cdot m_j), \tag{7}$$

where $P(i)$ represents the set of predecessor nodes for node $i$, and $m_j = w_v^\top h_j$ is the message passed from each predecessor. The embeddings are updated by aggregating these messages, with each message weighted by the attention coefficient $\alpha_j$.

## C  MODEL TRAINING

**Signal Probability Prediction (SPP)**  SPP is a classic and crucial task in the field of Boolean circuits representation learning Li et al. (2022); Shi et al. (2023; 2024); Liu et al. (2024), which is particularly important for tasks such as testability analysis, signal observability, and power estimation. We first compute probability that a logic gate outputs a logic 1 under random input simulation. To compute this, we perform a readout on the refined node-level embeddings and regress the predicted signal probability, as shown below

$$\hat{y}_i = \text{MLP}(t_i), \tag{8}$$

where $t_i$ represents the embedding vector of node $i$, and an MLP is used to predict the signal probability. Then, we compute the average absolute difference between the predicted signal probabilities $\hat{y}_i$, and the ground-truth probabilities $y_i$, which are measured using logic simulation with 15,000 random input patterns. Let $V$ denote the set of nodes in a training batch, where $N = |V|$ represents the total node count. The loss value is computed as:

$$L_{spp} = \frac{1}{N} \sum_{i \in \mathcal{V}} |y_i - \hat{y}_i|. \tag{9}$$

**Truth-Table Distance Prediction (TTDP)**  TTDP is a task for a more fine-grained evaluation of the functional representation capability of models. In the TTDP task, the embedding vectors of two nodes are considered similar if their corresponding node functions are similar. For nodes $i$ and $j$, we define their truth table vectors as $Z_i$ and $Z_j$, respectively, and the corresponding node tokens as $t_i$ and $t_j$, respectively. The functionality measurement can be expressed as

$$D^{token}(t_i, t_j) \propto D^{table}(Z_i, Z_j), \tag{10}$$

where the similarity between the functions for nodes $i$ and $j$ is measured based on the Hamming distance of their corresponding truth tables. Specifically, we compute the distance of the truth table by randomly sampling a sufficient number of truth table entries and calculating the Hamming distance for each node pair as follows

$$D^{table}(Z_i, Z_j) = \frac{\text{HammingDistance}(Z_i, Z_j)}{\text{length}(Z_i)}, \tag{11}$$

Table 6: Loss comparison between w/ multi-stage and w/o multi-stage.

|  | w/o multi-stage | w/ multi-stage | Red. ($\downarrow$) |
|---|---|---|---|
| $L_{spp}$ | 0.0231 | **0.0226** | 2.16% |
| $L_{ttdp}$ | 0.0809 | **0.0797** | 1.48% |

Then, the distance between the embedding vectors of node tokens $D^{token}(t_i, t_j)$, is expected to be proportional to the Hamming distance between their corresponding truth tables. To quantify this, we compute the distance between the embeddings using the cosine similarity as follows

$$D^{token}(t_i, t_j) = 1 - \frac{t_i^\top \cdot t_j}{\|t_i\| \cdot \|t_j\|}. \tag{12}$$

Finally, as the distance between the embedding vectors should be positively correlated with the actual truth table distance, the TTDP is calculated as follows, where $M$ is the number of sampled node pairs.

$$L_{ttdp} = \frac{1}{M} \sum_{(i,j) \in \mathcal{V}'} \left| \text{ZeroNorm}(D^{token}(t_i, t_j)) - \text{ZeroNorm}(D^{table}(Z_i, Z_j)) \right|. \tag{13}$$

**Loss Functions**   We incorporate multiple objectives into a three-stage curriculum learning procedure. In **Stage 1**, the model is trained with the probability prediction loss ($L_{spp}$) and the equivalence alignment loss ($L_{align}$), which encourages the model to capture signal distributions while aligning functionally equivalent nodes across views. In **Stage 2**, we introduce the functional prediction loss ($L_{ttdp}$), guiding the model to refine node embeddings toward accurate Boolean functionality while preserving alignment consistency. In **Stage 3**, we further add the multiview masked modeling loss ($L_{mcm}$), which requires the model to reconstruct masked cones using both intra-view and cross-view information.

This three-stage curriculum allows the model to gradually progress from learning low-level signal behavior, to capturing functional semantics, and finally to leveraging complementary multiview information for robust representation learning. The corresponding loss functions for each stage are defined in Eq. equation 14. In our experiments, we train for 60 epochs in Stage 1, 60 epochs in Stage 2, and 60 epochs in Stage 3.

$$\begin{aligned} L_{\text{stage1}} &= L_{\text{spp}} \cdot w_{\text{spp}} + L_{\text{align}} \cdot w_{\text{align}}, \\ L_{\text{stage2}} &= L_{\text{spp}} \cdot w_{\text{spp}} + L_{\text{align}} \cdot w_{\text{align}} + L_{\text{ttdp}} \cdot w_{\text{ttdp}}, \\ L_{\text{stage3}} &= L_{\text{spp}} \cdot w_{\text{spp}} + L_{\text{align}} \cdot w_{\text{align}} + L_{\text{ttdp}} \cdot w_{\text{ttdp}} + L_{\text{mcm}} \cdot w_{\text{mcm}}. \end{aligned} \tag{14}$$

where $w_{\text{spp}}, w_{\text{align}}, w_{\text{ttdp}}, w_{\text{mcm}}$ are the weighting coefficients for their respective loss terms.

To further validate the effectiveness of the multi-stage design, we compare the model trained with and without curriculum scheduling. As shown in Table 6, introducing multi-stage training consistently reduces both $L_{\text{spp}}$ and $L_{\text{ttdp}}$, with $L_{\text{ttdp}}$ achieving a relative reduction of $1.48\%$. Although the absolute gains are modest, the results indicate that progressive incorporation of objectives stabilizes optimization and leads to more refined functional representations.

## D   GENERALIZATION ANALYSIS ON OTHER ENCODERS

To assess MixGate's generalizability, we integrate its multiview fusion with several circuit encoders, evaluating each in two settings: (1) baseline setting (w/o MixGate), where the model operates using only a single-view circuit embedding, and (2) w/ MixGate, where additional circuit views are fused via the proposed framework.

Focusing on AIG refinement, results in Table 7 show consistent improvements across SPP and TTDP tasks. The **largest** and second-largest gains are highlighted. For instance, DeepGate3 Shi et al.

Table 7: Effect of MixGate with the other circuit encoders

| Encoder | w/o MixGate | | w/ MixGate | | | |
| | $L_{spp}$ | $L_{ttdp}$ | $L_{spp}$ | Red. ($\downarrow$) | $L_{ttdp}$ | Red. ($\downarrow$) |
| --- | --- | --- | --- | --- | --- | --- |
| GCN Kipf & Welling (2016) | 0.0419 | 0.1287 | 0.0298 | 28.88% | **0.0784** | **39.08%** |
| DeepGate2 Shi et al. (2023) | 0.0247 | 0.1156 | 0.0226 | 8.50% | 0.0797 | 31.11% |
| DeepGate3 Shi et al. (2024) | 0.0236 | 0.1054 | 0.0215 | 8.89% | 0.0722 | 31.50% |
| HOGA Deng et al. (2024) | 0.0641 | 0.2687 | **0.0440** | **31.36%** | 0.1685 | 37.29% |
| PolarGate Liu et al. (2024) | 0.074 | 0.1125 | 0.0623 | 15.81% | 0.0756 | 32.80% |

Table 8: MixGate performance on large-scale circuits

| Circuit | PI/PO | # AIG nodes | # AIG levels | $L_{spp}$ | $L_{ttdp}$ | Mem. (MB) | Time (s) |
| --- | --- | --- | --- | --- | --- | --- | --- |
| adder | 256/129 | 1,310 | 97 | 0.0215 | 0.0738 | 223 | 0.40 |
| arbiter | 256/129 | 4,589 | 14 | 0.0241 | 0.0722 | 760 | 1.20 |
| bar | 135/128 | 3,600 | 11 | 0.0242 | 0.0630 | 607 | 0.95 |
| cavlc | 10/11 | 600 | 10 | 0.0255 | 0.0740 | 105 | 0.20 |
| ctrl | 7/26 | 85 | 6 | 0.0259 | 0.0747 | 17 | 0.10 |
| dec | 8/256 | 304 | 3 | 0.0257 | 0.0745 | 53 | 0.10 |
| div | 128/128 | 58,798 | 1,621 | 0.0205 | 0.0640 | 9,596 | 14.72 |
| i2c | 147/142 | 1,013 | 8 | 0.0248 | 0.0730 | 170 | 0.30 |
| int2float | 11/7 | 208 | 9 | 0.0256 | 0.0743 | 39 | 0.07 |
| log2 | 32/32 | 38,817 | 202 | 0.0225 | 0.0670 | 6,403 | 10.06 |
| max | 512/130 | 4,378 | 29 | 0.0240 | 0.0720 | 721 | 1.10 |
| mem_ctrl | 1,204/1,231 | 33,661 | 34 | 0.0228 | 0.0680 | 5,503 | 8.50 |
| multiplier | 128/128 | 31,817 | 125 | 0.0232 | 0.0690 | 5,217 | 8.00 |
| priority | 128/8 | 457 | 11 | 0.0252 | 0.0735 | 84 | 0.12 |
| router | 60/30 | 159 | 12 | 0.0256 | 0.0742 | 32 | 0.06 |
| sin | 24/25 | 6,896 | 100 | 0.0259 | 0.0718 | 1,137 | 1.80 |
| sqrt | 128/64 | 30,670 | 1,637 | 0.0222 | 0.0675 | 5,054 | 8.00 |
| square | 64/128 | 17,405 | 112 | 0.0235 | 0.0705 | 2,851 | 4.50 |
| voter | 1,001/1 | 9,375 | 48 | 0.0231 | 0.0695 | 1,553 | 2.50 |

(2024) sees SPP loss drop by 8.89% and TTDP by 31.50%. GCN Kipf & Welling (2016) and HOGA Deng et al. (2024) benefit most, where GCN reduces losses by 28.88% ($L_{spp}$) and 39.08% ($L_{ttdp}$), HOGA by 31.36% and 37.29%. As both models focus on local structures, MixGate's multiview signals provide crucial complementary global context.

# E  GENERALIZATION ANALYSIS ON LARGE-SCALE CIRCUITS

To validate the generalization ability of MixGate on large-scale circuits, we conduct experiments with the circuits in EPFL benchmarks Amarú et al. (2015)[3]. As shown in Table 8, MixGate maintains strong performance across key metrics even for circuits with over 30,000 original nodes and 1,600 logic levels (e.g., *div*, *sqrt*), demonstrating remarkable scalability.

Notably, the $L_{spp}$ and $L_{ttdp}$ metrics remain in small ranges of 0.0205–0.0259 and 0.0630–0.0747, respectively, which align with small circuits. For instance, the *div* circuit (58,798 nodes, 1,621 levels) achieves an $L_{spp}$ of 0.0205 and $L_{ttdp}$ of 0.0640 with only 9,596 MB memory usage, while the *sqrt* circuit (30,670 nodes, 1,637 levels) attains $L_{spp}$=0.0222 and $L_{ttdp}$=0.0675 using 5,054 MB memory. This linear growth in resource consumption relative to circuit size confirms the computational efficiency of our model. Furthermore, the runtime scales sublinearly – for circuits like *multiplier* (31,817 nodes) and *mem_ctrl* (33,661 nodes), MixGate completes optimization in 8.00s and 8.50s, respectively, showcasing its practical viability for industrial-scale applications. These results systematically prove that MixGate preserves model performance while avoiding exponential complexity growth.

---

[3] The EPFL Combinational Benchmark Suite, https://github.com/lsils/benchmarks

Table 9: Contribution of Each View

| Experiment | Modal Composition | $L_{spp}$ | Inc.($\uparrow$) | $L_{ttdp}$ | Inc.($\uparrow$) |
|---|---|---|---|---|---|
| Exp-0 | AIG, MIG, XAG, XMG | 0.0220 | - | 0.0737 | - |
| Exp-1 | AIG, MIG, XAG | 0.0232 | 5.45% | 0.0881 | 19.54% |
| Exp-2 | AIG, MIG, XMG | 0.0229 | 4.09% | 0.0847 | 14.93% |
| Exp-3 | AIG, XAG, XMG | 0.0226 | 2.73% | 0.0796 | 8.01% |
| Exp-4 | AIG | 0.0240 | 9.09% | 0.1134 | 53.87% |

Table 10: Transformer Hyperparameters

| Parameter | Value |
|---|---|
| Hidden Dimension | 128 |
| Number of Heads | 8 |
| Number of Layers | 2 |
| FFN Hidden Dimension | 128 |
| Dropout Rate | 0.1 |
| Residual Connections | yes |
| Layer Normalization | yes |
| Transformer Encoder Layers | 2 |

## F CONTRIBUTION OF EACH VIEW

In this section, we analyze the contribution of each modality by conducting ablation experiments, where each experiment removes one of the graph views from the multiview fusion process. We evaluate the impact of each view using the SPP and TTDP on the AIGs, serving as the refinement target. All experiments are conducted under the same settings as described in the previous sections to ensure fair comparison.

We report the ablation results for each graph view by measuring SPP and TTDP in Table 9, along with their increases (Inc.) compared to the full multiview setup in Exp-0. Removing the XMG view in Exp-1 leads to a 5.45% increase in SPP and a 19.54% increase in TTDP, indicating that XMG provides highly complementary information to AIG, especially in terms of capturing semantically aligned logic. In Exp-2, when XAG is removed, the SPP and TTDP values rise by 4.09% and 14.93%, respectively, showing its moderate contribution. Exp-3, which excludes MIG, shows the smallest degradation: 2.73% in SPP and 8.01% in TTDP. Finally, Exp-4, which uses only AIG, suffers the most, with over 9% degradation in SPP and nearly 54% degradation in TTDP, confirming the importance of multiview fusion.

These results suggest that XMG offers the most complementary view to AIG, likely due to its inclusion of both XOR and MAJ gates, which enrich the semantic space. In contrast, MIG shares greater structural similarity with AIG, and thus contributes less unique information, explaining its smaller impact when removed.

## G MODEL IMPLEMENTATION

**Hyperparameters** The Transformer model, detailed in Table 10, uses a hidden dimension of 128, with 4 attention heads, 2 layers, and a dropout rate of 0.1. The GNN encoder, detailed in Table 11, also utilizes a hidden dimension of 128, paired with an MLP for update with a hidden dimension of 32, operating in 1 round with a batch size of 32 and a learning rate of $10^{-4}$. The hierarchical tokenizer, as outlined in Table 12, is configured with 8 pooling transformer heads, 2 layers. It incorporates a graph hierarchy with a maximum of 128 nodes per hop and 5 hops per subgraph, while the graph partitioning process utilizes the parameters $k$ (maximum level) and $q$ (stride) to manage subgraph size and overlap.

**Training Environment** We conduct all model training using 8 NVIDIA A800-SXM4-80GB GPUs. The graph encoders for AIG, MIG, XAG, and XMG are trained independently for 120

Table 11: GNN Encoder Hyperparameters

| Category | Parameter | Value |
|---|---|---|
| Aggregator | Number of Rounds | 1 |
| | Hidden Dimension | 128 |
| | MLP Hidden Dimension | 32 |
| Readout | MLP Layers | 3 |
| | Readout Dropout | 0.2 |
| | Normalization Layer | BatchNorm |
| | Activation Function | ReLU |
| Training | Learning Rate | $10^{-4}$ |
| | Batch Size | 256 |
| | Optimizer | Adam |

Table 12: Hierarchical Tokenizer Hyperparameters

| Category | Parameter | Value |
|---|---|---|
| Token Generation | CLS Token Init Method | Random initialization |
| Hierarchical Pooling | Pooling Transformer Heads | 8 |
| | Pooling Transformer Layers | 2 |
| | FFN Hidden Multiplier | 4 |
| Graph Hierarchy | Max Nodes per Hop | 128 |
| | Max Hops per Subgraph | 5 |
| Graph Partitioning | $k$ (Maximum Level) | 8 |
| | $q$ (Stride) | 8 |

epochs with a batch size of 256, ensuring sufficient convergence in loss. After pre-training, we proceed to refine the AIG embeddings using the multiview framework for an additional 120 epochs with a batch size of 128. This procedure is applied symmetrically when refining other representations.

## H COMPARISON ON EQUIVALENCE ALIGNMENT LOSS

**Contrastive Formulation** For completeness, we also experimented with a contrastive alignment objective of the InfoNCE form:

$$L_{\text{contrast}} = - \sum_{(i,j) \in \mathcal{P}} \log \frac{\exp(\text{sim}(hf_i, hf_j)/\tau)}{\exp(\text{sim}(hf_i, hf_j)/\tau) + \sum_{k \in \mathcal{N}(i)} \exp(\text{sim}(hf_i, hf_k)/\tau)}, \quad (15)$$

where $hf_i, hf_j, hf_k$ denote the embeddings of nodes $i, j, k$ from different circuit views, $\text{sim}(\cdot, \cdot)$ is cosine similarity, $\tau$ is a temperature hyperparameter, $\mathcal{P}$ is the set of equivalent (positive) node pairs, and $\mathcal{N}(i)$ is the set of negatives sampled for anchor $i$.

**Discussion** We chose the L1 distance as the alignment loss due to its simplicity, robustness, and suitability to circuit data. Unlike images or text, circuit netlists exhibit strong structural heterogeneity: two functionally equivalent nodes may reside in drastically different local neighborhoods. The critical challenge is thus to provide a stable anchor for positive pairs rather than to repel negatives. L1 loss directly optimizes this objective and avoids the large-scale negative sampling required by contrastive learning, which becomes prohibitively expensive on circuit graphs. Moreover, subsequent masked modeling and downstream tasks already inject rich discriminative supervision, reducing the necessity of additional contrastive terms.

**Results** As shown in Table 13, the contrastive variant offers no improvement in $L_{spp}$ or $L_{ttdp}$, but incurs nearly $1.5\times$ higher memory usage and runtime. This indicates that more complex objectives bring significant computational overhead without clear accuracy benefits, while the L1 design achieves better stability–efficiency trade-offs.

Table 13: L1 Loss vs Contrastive Loss

| Method | L1 Loss | | | | Contrastive Loss | | | |
|---|---|---|---|---|---|---|---|---|
| | $L_{spp}$ | $L_{ttdp}$ | Mem. (MB) | Time (s) | $L_{spp}$ | $L_{ttdp}$ | Mem. (MB) | Time (s) |
| AIG | 0.0226 | 0.0797 | **8,674.1** | **7.16** | 0.0229 | 0.0801 | 13,121.3 (+51.3%) | 10.78 (+50.6%) |

## I CONTRIBUTION STATEMENT

We devoted approximately 336 hours to constructing a multiview circuit dataset and generating fine-grained equivalence alignment labels based on the open-source ForgeEDA dataset. In the future, we will release both our dataset and source code to the AI and EDA communities, aiming to foster open research and collaboration. We firmly believe that MixGate will play a significant role in advancing research on multiview circuit representation learning and unsupervised learning in chip design.

## J LIMITATIONS AND FUTURE WORK

This study focuses on different graph-based representations derived from circuit netlists, where we can obtain node-level equivalence labels via SAT-based checks. While the alignment-first principle and MixGate's curriculum generalize in spirit, applying them across more heterogeneous EDA artifacts will require new mechanisms for establishing fine-grained cross-modal correspondences. A natural next step is to explore alignment methods that bridge semantic gaps between these abstraction levels, and to investigate scalable approximations for equivalence discovery in very large designs.

## K THE USE OF LARGE LANGUAGE MODELS (LLMS)

In this paper, large language model (LLM) is employed as an assistive technology to improve the clarity and quality of the writing. The use of the LLM is strictly limited to polishing the language, grammar, and formatting of the text. The model does not contribute to, generate, or interpret any of the core scientific ideas, data analysis, or conclusions presented in this work.

