# OpenReview forum: "Alignment Unlocks Complementarity: A Framework for Multiview Circuit Representation Learning"
_ICLR.cc/2026/Conference — ICLR 2026 Conference Withdrawn Submission_

### Official Review · Reviewer_S5aC · 2025-10-23

**Soundness:** 3
**Presentation:** 1
**Contribution:** 3
**Rating:** 4
**Confidence:** 3

**Summary:**

This paper addresses the challenge of multiview learning on Boolean circuits, where structural heterogeneity between representations (e.g., AIG and XMG) hinders effective fusion. The authors propose MixGate, a framework that first aligns functionally equivalent views through an Equivalence Alignment Loss before applying multiview masked modeling. Experiments and ablations show that this alignment-first curriculum significantly enhances the effectiveness of self-supervised learning across heterogeneous circuit views.

**Strengths:**

- The ablation study is comprehensive.
- The design is effective.

**Weaknesses:**

- The design is complicated and lacks details.
- Comparison with baselines is missing.
- Experiments are not comprehensive.
- There are performance inconsistency.

**Questions:**

* One of the main motivations is that "the cross-view context is perceived as noise rather than a useful signal". Are there any theoretical or empirical analyses supporting this claim?
* What is the rationale for introducing both structural and functional embeddings in the graph encoder? How do they complement each other?
* Figure 1 does not illustrate the dual embedding design; please clarify the architecture visually.
* An ablation study using only one embedding type (either structural or functional) is missing.
* What is the pretraining strategy for the graph encoders? Is it part of the proposed three-stage training scheme?
* The training strategy described in Training Environment appears different from the proposed three-stage strategy. How do these two strategies relate or work together?
* Are the encoders shared among views, or does each view use its own encoder? For instance, is $\Phi_{AND}$ shared across all four views or not?
* What is the motivation behind the Hierarchical Circuit Tokenizer?
* How are the two types of embeddings integrated or fed into the tokenizer?
* Please clarify the definition of $\mathcal{G}^S$ in Line 323.
* Provide dataset statistics for training, validation, and test splits to enable reproducibility.
* The absence of baseline comparisons makes it difficult to evaluate the claimed superiority; please compare with SOTA graph self-supervised learning methods.
* Include convergence curves for all four loss components to better illustrate optimization behavior.
* The results of $L_{mcm}$ and $L_{align}$ in Table 2 are inconsistent with those in Table 1.
* What is the average actual mask ratio used during masked modeling?
* The results of $L_{spp}$ and $L_{ttdp}$ in Table 9 also appear inconsistent with Table 1.
* Hyperparameter sensitivity analyses are missing, including fan-in depth $l$ in tokenizer, $\mathcal{P}$ in Eq.4, $M$ in Eq.13, $w_{spp}$, $w_{align}$, $w_{ttdp}$, $w_{mcm}$ in Eq.14, the parameters $k$ (maximum level) and $q$ (stride) to manage subgraph size and overlap.
* The definition of the truth-table distance $D^{table}(Z_i, Z_j)$ in Eq.11 is asymmetric, as the denominator uses only $\text{length}(Z_i)$. This design choice violates the natural symmetry of pairwise distance metrics, i.e., $D^{table}(Z_i, Z_j) \neq D^{table}(Z_j, Z_i)$. Since the token distance $D^{token}(t_i, t_j)$ is symmetric by definition, the inconsistency introduces conflicting supervision signals in the training objective $L_{ttdp}$. Consequently, the optimization target becomes ill-defined, and the learned embeddings may reflect artifacts from the table length rather than true relational similarity.

---

### Official Review · Reviewer_2rvM · 2025-10-27

**Soundness:** 2
**Presentation:** 2
**Contribution:** 2
**Rating:** 4
**Confidence:** 3

**Summary:**

This paper proposes the MixGate framework to address the structural heterogeneity in multiview learning for Boolean circuits. The key insight is to first perform functional alignment across different circuit representations (e.g., AIG, XMG, MIG) to bridge the structural differences, followed by multiview masked modeling (MCM) to improve performance.

**Strengths:**

The MixGate framework not only works for current circuit graph representations but also has scalability to other encoders and tasks.

**Weaknesses:**

1.While the method improves performance, it significantly increases memory and runtime costs, especially when handling multiple views and complex alignment methods.

2.Although scalable, the method may face performance bottlenecks when handling very large circuit designs, especially during multiview data processing.

3. The method heavily relies on accurate functional alignment labels, which may be time-consuming and difficult to generate, limiting its widespread use in practical applications.

4. While the paper focuses on multiview modeling, there is a lack of comparison with other self-supervised learning methods, which could provide further insight into the superiority of the proposed approach.

**Questions:**

1.For larger-scale circuit designs, are there plans to develop more efficient algorithms to handle multiview data?

2.In practical deployment, how can the accuracy of functional alignment be ensured, especially for large-scale circuits?

---

### Official Review · Reviewer_u5aw · 2025-10-27

**Soundness:** 2
**Presentation:** 2
**Contribution:** 2
**Rating:** 2
**Confidence:** 4

**Summary:**

This work introduces MixGate, a framework built on a principled training curriculum that first teaches the model a shared, function-aware representation space via an Equivalence Alignment Loss.

It identifies structural heterogeneity as a critical barrier for self-supervised learning on circuits and proposes a novel alignment-first curriculum as a solution. Then they present MixGate, a complete and effective framework embodying their alignment-first principle.

**Strengths:**

1. This work proposed to align different netlist formats/views (AIG, MIG, etc) at a fine-grained granularity, which is on equivalent nodes. Previous works do not perform the alignment at the node level.
2. To generate labels for functionality alignment, this work proposed SAT-based solutions to verify node equivalence.

**Weaknesses:**

1. Lack of realistic application for validation. In the experiment, this work mostly evaluates its solution using the training loss of the proposed learning loss functions. Seems there is even no testing set during the experiment. Achieving a lower value on this work's own proposed loss cannot indicate a better methodology, especially since this loss seems to be calculated based on the training dataset. This work does not elaborate on which application will benefit from this solution. Due to this problem, it is also not very clear whether the main target of this work is to generate an embedding of each node or the whole circuit.

2. Lack of prior works as baselines. Since this work is evaluated on its own loss scores, no prior work has been compared with it in the main content. There have been many prior works, as introduced in this paper.

3. Limited novelty. The overall graph learning method is quite similar to prior works, such as DeepGate2. There have also been multiple works on learning multiple different netlist structures with the same functionality. The reviewer believes the main novelty of this work is the "fine-grained" alignment across different "views" at the node level. But this part alone may not be significant enough. In addition, this technique may be difficult to implement due to a high cost, as mentioned in the next point.

4. The cost of "Equivalence Gates Identification" is not sufficiently elaborated. The cost seems to be quite high if scaling up to larger designs. The technique of "Equivalence Gates Identification" requires SAT checking, an NP-complete algorithm. On larger circuits, when each node is far away from the PI/registers, checking the equivalence could be extremely slow.

**Questions:**

Please see the weakness part.

---

### Official Review · Reviewer_EEhN · 2025-10-31

**Soundness:** 3
**Presentation:** 3
**Contribution:** 2
**Rating:** 6
**Confidence:** 4

**Summary:**

The paper “Alignment Unlocks Complementarity: A Framework for Multiview Circuit Representation Learning” presents MixGate, a framework for multiview self-supervised learning on Boolean circuits. It addresses the challenge that different circuit graph representations, such as And Inverter Graphs (AIGs), Majority Inverter Graphs (MIGs), XOR AND Graphs (XAGs), and XOR Majority Graphs (XMGs), although functionally equivalent, have very different structures. This structural heterogeneity makes it difficult for models to effectively combine information from multiple views using self-supervised techniques like masked circuit modeling. The authors argue that before such techniques can succeed, the model must first learn to align these structurally diverse views at a functional level.

The main contribution is the alignment first principle, which proposes that the model should first learn a shared, function aware embedding space across different circuit formats before applying multiview self-supervision. To achieve this, the authors introduce the Equivalence Alignment Loss, which enforces functional consistency among nodes that are verified to be equivalent across views using SAT based checking. This encourages the model to produce similar embeddings for functionally identical components, regardless of their structural representation.

The proposed MixGate framework integrates several key components: multiple specialized graph encoders for each circuit view, a hierarchical circuit tokenizer that organizes embeddings into hop, subgraph, and graph level tokens, and a Transformer based fusion module that learns to combine multiview information effectively. Once alignment is achieved, a multiview masked modeling objective is applied, allowing the model to benefit from complementary structural and semantic information across views.

Extensive experiments on the ForgeEDA dataset demonstrate that MixGate substantially improves performance on circuit representation learning tasks. The results show that alignment is essential for unlocking the benefits of multiview self-supervision, transforming masked modeling from an ineffective technique into a major performance driver.

**Strengths:**

Originality:
The paper presents a genuinely original perspective on multiview circuit representation learning. While previous works in circuit learning have explored single view encoders or naive multiview fusion, this paper introduces the alignment first principle, arguing that functional alignment is a prerequisite for effective self supervised learning across heterogeneous circuit views. The formulation of the Equivalence Alignment Loss, which enforces function level consistency across structurally distinct representations, is a conceptually novel and technically elegant idea. The framework also creatively combines existing components such as graph encoders, tokenization, and Transformers into a cohesive and principled training curriculum that redefines how complementary circuit views can be learned jointly.

Quality:
The technical quality of the work is high. The methodology is sound and builds on well established tools from logic synthesis and circuit verification such as SAT sweeping to ensure rigorous alignment labeling. The experimental design is comprehensive and includes clear ablation studies that isolate the effects of alignment and masking, parameter sensitivity analyses such as mask ratios, and comparisons across multiple circuit formats including AIG, MIG, XAG, and XMG. The results consistently support the central hypothesis that alignment unlocks the potential of multiview masked modeling. The integration of hierarchical tokenization for efficiency and scalability further demonstrates careful engineering and thoughtful design.

Clarity:
The paper is clearly written and well structured, making a complex technical topic accessible. The motivation is articulated early, with clear illustrations that show how different circuit representations encode complementary information. The framework is explained step by step, and visual aids such as Figures 1 to 4 effectively convey both the system design and the data flow. The inclusion of detailed appendices enhances transparency and reproducibility. While some sections could be more concise, the overall presentation maintains a good balance between intuition, formalism, and implementation detail.

Significance:
This work makes a meaningful contribution to the emerging area of self supervised and multiview learning for electronic design automation. It addresses a long standing limitation, namely how to leverage the complementarity of different circuit representations without being hindered by their structural differences. By establishing alignment as a prerequisite for cross view learning, the paper provides a general and transferable principle that can extend beyond circuit modeling to other domains with heterogeneous graph structures. The empirical improvements are substantial, and the framework’s modular design suggests that it could easily integrate with future circuit encoders and industrial design tools. Overall, the paper offers both conceptual and practical significance and is likely to influence subsequent work in representation learning for hardware design.

**Weaknesses:**

Limited exploration of scalability and practical applicability:
Although the paper presents convincing evidence of MixGate’s effectiveness on the ForgeEDA dataset, it does not fully examine the scalability of the proposed alignment process. The Equivalence Alignment Loss depends on SAT-based equivalence checking, which can become computationally expensive for large industrial circuits with millions of nodes. The paper would benefit from empirical runtime or memory analyses that show how MixGate scales with circuit size and whether approximation or sampling strategies could maintain alignment quality under practical constraints.

Narrow experimental scope and limited downstream validation:
Most evaluations focus on pretraining metrics such as Structural Property Prediction (SPP) and Truth Table Distribution Prediction (TTDP). While these are informative, they are still proxy tasks. It remains unclear how well the learned representations transfer to real downstream applications like circuit optimization, logic synthesis, or power-delay estimation. Including such downstream tasks would more directly validate the practical utility of the proposed framework and strengthen the claim that alignment improves general circuit understanding.

Dependence on manually defined circuit views:
MixGate currently operates on a fixed set of four predefined circuit representations (AIG, MIG, XAG, XMG). This design choice, while effective, limits flexibility and raises questions about extensibility. Could the framework automatically discover or adapt to new circuit abstractions rather than relying on manually selected ones? The authors could explore whether learned or dynamically generated views might further improve complementarity or reduce redundancy between existing ones.

Limited interpretability of the learned alignment:
Although the paper claims that aligned embeddings capture functionally consistent behavior, the work does not provide clear qualitative or quantitative analyses to illustrate this. Visualizations such as embedding similarity maps, cross-view attention distributions, or case studies showing how alignment improves specific circuit examples would help substantiate this claim. These analyses would also make the model more interpretable and support the assertion that alignment meaningfully enhances representation quality.

Clarification needed on hierarchy design choices:
The hierarchical circuit tokenizer organizes information across hop, subgraph, and graph levels, but the rationale for this specific structure is not fully justified. It is unclear whether these levels were empirically chosen or theoretically motivated. A comparison with simpler or alternative hierarchies (for example, learned or adaptive ones) would clarify whether the performance gains come from the hierarchy itself or from the alignment mechanism.

**Questions:**

Scalability of alignment supervision:
The proposed Equivalence Alignment Loss depends on SAT-based equivalence checking to identify functionally equivalent nodes across circuit views. Could the authors clarify the computational cost and scalability of this process on large industrial circuits? Is there a feasible way to approximate or batch this alignment to reduce runtime while maintaining accuracy? Empirical data or complexity analysis on larger circuits would be helpful.

Practical relevance of pretraining objectives:
The experiments primarily focus on SPP and TTDP as pretraining tasks. Could the authors discuss how improvements in these proxy metrics translate to real-world circuit design or optimization tasks, such as power estimation, timing analysis, or logic synthesis? A small-scale experiment or discussion of transferability would strengthen the practical impact of the framework.

Flexibility to new circuit representations:
MixGate currently handles four manually chosen circuit formats (AIG, MIG, XAG, XMG). How flexible is the method to new or hybrid circuit representations that might not have one-to-one functional mappings with these formats? Could the alignment-first principle extend to views that differ more substantially in abstraction level, such as HDL-level or gate-level representations?

Interpretability of aligned embeddings:
One of the main claims is that alignment encourages functionally consistent embeddings across structurally different circuits. Could the authors provide empirical or visual evidence of this behavior, for instance, through similarity plots, attention maps, or qualitative examples showing that aligned nodes correspond to similar logical roles across views? This would make the alignment mechanism more transparent and convincing.

Design motivation for hierarchical tokenization:
The hierarchical circuit tokenizer aggregates information across hop, subgraph, and graph levels. Could the authors elaborate on the design rationale behind this hierarchy? Was it chosen empirically or theoretically motivated by circuit properties? Have the authors tested alternative tokenization strategies or adaptive hierarchies that might better capture structural variations in large circuits?

---

### Note · Authors · 2025-11-18

I have read and agree with the venue's withdrawal policy on behalf of myself and my co-authors.